# Determinants of low birth weight among newborns delivered in public hospitals of North Shewa Zone, Amhara region, Ethiopia: A case-control study (2023)

**Beniyas Minda[1], Girma Bekele[2], Solomon Hailemeskel[3], Abera Lambebo[1]***

1 Debre Berhan University Asrat Woldeyes Health Science Campus School of Public Health Department of Nutrition, Debre Berhan, Ethiopia, 2 Debre Berhan University Asrat Woldeyes Health Science Campus School of Public Health Department of Epidemiology, Debre Berhan, Ethiopia, 3 Debre Berhan University Asrat Woldeyes Health Science Campus School of Nursing Department of Midwifery, Debre Berhan, Ethiopia

* lambebo70@gmail.com

## Abstract

### Background

Low birth weight (LBW), defined as a birth weight less than 2500 g, irrespective of gestational age, poses a significant health concern for newborns. Despite efforts, the incidence of LBW in sub-Saharan Africa has remained stagnant over the past decade, warranting attention from healthcare providers, policymakers, and researchers.

### Objective

This study aimed to identify factors associated with LBW among newborns delivered in public hospitals of North Shewa Zone, Amhara Region, Ethiopia, from May 2 to June 10, 2023.

### Methods and materials

An unmatched case-control study was conducted from May 2 to June 10, 2023, involving 318 participants (106 cases and 212 controls). Data were collected using pretested interviewer-administered structured questionnaires, medical record reviews, and direct anthropometric measurements. Bivariate analyses were conducted, and variables with a p-value $\leq 0.25$ were included in a multivariable logistic regression model to determine significant determinants of LBW. A significance level of $p < 0.05$ was used.

### Results

A total of 309 newborns (103 cases and 206 controls) were included, yielding a response rate of 97.2%. Among the findings, females exhibited a higher risk of LBW (adjusted odds ratio [AOR]: 3.13, 95% CI: 1.34, 7.32, p = 0.008), as did mothers aged 20 or younger (AOR: 3.42, 95% CI: 1.35, 8.66, p = 0.009). Lack of formal education was associated with increased risk (AOR: 6.82, 95% CI: 2.94, 15.3, p < 0.001), as were unplanned pregnancies (AOR: 3.08, 95% CI: 1.38, 6.84, p = 0.006) and missed antenatal care visits (AOR: 2.74,

**Funding:** The author(s) received no specific funding for this work.

**Competing interests:** The authors have declared that no competing interests exist.

95% CI: 1.16, 6.49, p = 0.021). No significant associations were found with residency type or maternal age above 35.

## Conclusion

Mothers aged ≤ 20 years, with inadequate minimum dietary diversity, lack of antenatal care attendance, and unplanned pregnancies, faced heightened risks of LBW. Addressing these factors is vital for reducing LBW occurrences and improving newborn health outcomes in Ethiopia.

## Introduction

### Background

Low birth weight (LBW) is a significant public health concern worldwide, particularly in developing countries like Ethiopia. LBW refers to a birth weight less than 2,500 grams (5.5 pounds) regardless of gestational age, as defined by the World Health Organization (WHO) [1].

Global low birth weight (LBW) incidence is estimated at 15–20%, exceeding 20 million births annually, with 95.6% occurring in low- and middle-income countries. Regional variations include 28% in South Asia, 13% in Sub-Saharan Africa, and 9% in Latin America [2]. In Ethiopia, LBW prevalence varies across regions and settings, ranging from 8% in the 2011 Ethiopian Demographic and Health Survey (EDHS) to 20.2% in a rural Ethiopia study in 2016 [3].

LBW stems from preterm delivery and intrauterine growth restriction. The unclear etiology of preterm delivery includes factors like hypertension, malaria, syphilis, and HIV [4]. In Ethiopia, rural areas pose higher LBW risks, tied to factors such as limited antenatal care, reliance on firewood, insufficient iron and folic acid intake, maternal under-nutrition, height, neonate sex, lack of nutrition counseling, insufficient snacks during pregnancy, maternal anemia, and inadequate dietary diversity [3,5–8]. Preconception and first-trimester anemia link to increased LBW, preterm birth, and neonatal mortality [8,9].

Low birth weight (LBW) has lasting health impacts, correlating with childhood diseases, non-communicable conditions like diabetes and cardiovascular issues, and intergenerational consequences [9,10]. In Ethiopia, a high neonatal mortality rate of 30 deaths per 1,000 live births, reported in the 2019 Ethiopian Demographic and Health Survey, is largely attributed to LBW [11]. Implemented strategies include improving birth spacing, prenatal care, infection prevention, and managing hypertension during pregnancy [12,13]. However, achieving the global nutrition target of a 30% reduction in LBW by 2025 requires more than doubling current reduction rates [2].

Despite extensive efforts, LBW remains a significant public health challenge in Ethiopia, particularly in the North Shewa Zone, where the absence of institution-based case-control studies highlights the need for context-specific research to identify LBW factors and inform targeted interventions effectively.

## Methods and materials

A hospital-based unmatched case-control study was conducted from May 2 to June 10, 2023, in public hospitals within the North Shewa Zone, Amhara region, Ethiopia. The study area, located approximately 130 km Northeast of Addis Ababa, is characterized by a dense population and rapid urbanization, divided into twenty-two administrative districts known as

Woredas. As of 2022, the North Shewa Zone has a total population of 2,405,537, comprising 1,178,713 females and 1,226,825 males, according to Ethiopian population projections.

Among the 11 governmental public hospitals in the North Shewa Zone, four were selected through simple random sampling for the study. These included 1 comprehensive, 1 teaching and 2 general hospitals.

**Source Population:** The source population comprised all mothers with their newborns delivered at the selected governmental public hospitals of North Shewa Zone.

**Study Population**: The study population included mothers with their newborns with birth weight less than 2500 grams (cases) and mothers with newborns with birth weight between 2500 grams to 4000 grams (controls) delivered during the data collection period at the selected hospitals.

**Sample Size Determination**: The sample size for cases and controls was calculated using Epi Info version 7.2.5 software. The required sample size was 318 (106 cases and 212 controls), with a case-to-control ratio of 1:2. The sample size calculation considered a proportion of control (Maternal Age $\leq 18$) as 11.6%, a proportion of cases as 21.4%, an adjusted odds ratio of 2.69, a study power of 80%, a 95% confidence interval, and a non-response rate of 15%.

**Sampling Technique and Procedure:** In the hospital-based unmatched case-control study conducted in the North Shewa Zone, Amhara region, Ethiopia, four hospitals were chosen through simple random sampling. The distribution of the required sample size was proportionally assigned to each hospital, considering the reported number of deliveries from the year before the survey. Cases were then selected purposely and for every case, two controls were chosen through systematic random sampling.

**Data Collection Method**: Data for the study was gathered from the selected hospitals through a structured questionnaire, covering a range of factors including socio-demographics, maternal behavior, anthropometric measurements, antenatal care, gestational age, weight gain, and relevant medical history. BSc midwifery professionals, experienced in the selected hospitals, collected the data under the supervision of public health professionals. Informed consent was obtained from each hospital's CEO before the commencement of data collection.

Maternal and neonatal data were collected within the first 6 hours post-delivery. The measurements were taken using standard tools: birth weight was measured with a balanced seca scale, maternal Mid-Upper Arm Circumference (MUAC) with a non-stretchable MUAC tape, and maternal height and weight with a wall height scale and beam balance, respectively. Interviews were conducted with all mothers who had singleton live births during the data collection period at the selected hospitals, encompassing both controls and cases. Additionally, client charts were reviewed for information not covered during the interview process, ensuring a comprehensive and accurate dataset.

## Ethical consideration

Before commencing the study, the research team obtained appropriate ethical clearance and a supportive letter from the Ethical Review Committee of Aserat Woldeyes College of Health Science. Written permission was also obtained from the selected hospitals to conduct the study on their premises.

Informed Consent: Written informed consent was obtained from all mothers who participated in the study or their caregivers, ensuring that they were fully aware of the purpose, procedures, potential risks, and benefits of the study. Participants were informed that their involvement was voluntary, and they had the right to withdraw from the study at any point without facing any negative consequences.

Confidentiality: To protect the privacy and confidentiality of the participants, all provided information was treated with the utmost confidentiality. Personal identifiers such as names were not recorded, and a unique code number was used to link the data to each participant. The collected data was securely stored and accessible only to the research team, maintaining the anonymity of the participants.

Additionally, the data collectors were trained in ethical research practices to ensure that the participants' rights and well-being were respected throughout the study. study's aim, purpose, benefit, risk, discomfort, and right to refuse or withdraw at any time for any reason by applying Helensic principles [14].

## Result

### Socio-demographic and socio-economic characteristics of study participants

A total of 309 (103 cases and 206 controls) newborns were included in this study with a response rate of 97.2%. Forty-nine (15.86%) of the mothers of the cases and 154 (49.84%) of controls were found in the age group 21–34 years. The median age of the participants was 24 ±7. The majority of the respondents of cases (71.84%) did not attend formal school, while most mothers of the controls (78.1%) attended formal school. The husbands of the majority of respondents of cases (64.07%) did not attend formal education, whereas 73.78% of controls attended formal education (Table 1).

### Obstetric/Reproductive and medical-related characteristics of respondents

Most mothers of the controls (95.14%) and cases (93.20%) had no history of abortion. Similarly, the majority of mothers of controls (95.14%) and cases (30.74%) had no history of stillbirth. Most of the newborns' sex in the control group (73.30%) were male, while most of the newborns' sex in the case group (65.04%) were female. The pregnancy in 67.04% of cases was not planned, whereas 77.66% of controls had wanted and planned pregnancies (Table 2).

### ANC service utilization and nutrition-related characteristics of study participants

From the total respondents, 66.01% of cases did not attend ANC, while 65.53% of controls attended ANC services. About 26.21% of the cases had ≤3 ANC visits, whereas 82.52% of the controls had ≥4 ANC visits during their pregnancy. Among the total respondents, 53.39% of cases had MUAC ≤23cm, while 17.96% of controls had MUAC ≤23cm (Table 3).

### MDDS-W consumption rate of ten food groups on a daily basis

Among the total respondents, 23.62% of mothers of cases had eaten food pulses, whereas 61.48% of mothers of the controls had eaten food pulses within 24 hours. Overall, 68.61% of respondents fulfilled the minimum dietary requirements by consuming at least five out of ten defined food groups in the previous day or night (Fig 1).

### Behavioral-related factors which determine LBW

Among the total respondents, 62.13% of cases had indoor cooking history, whereas 71.35% of controls had no history of indoor cooking. Regarding coffee consumption, 44.07% of cases reported drinking coffee, whereas 65.04% of controls had no history of coffee drinking during

**Table 1. Socio demographic and socio-economic characteristics of mothers who gave birth at governmental public hospitals of North Shewa Zone, 2023 (cases = 103, controls = 206).**

| Variables | Category | Birth weight | | |
| --- | --- | --- | --- | --- |
| | | Cases (n = 103) Frequency (%) | Controls(n = 206) Frequency (%) | Total (%) |
| Age of mother | ≤20 | 41(13.26) | 23(7.44) | 64(20.71) |
| | 21–34 | 49(15.85) | 154(49.83) | 203(65.70) |
| | ≥35 | 13(4.2) | 29(9.38) | 42(13.59) |
| Educational status of mother | No attend formal school/education | 74(23.95) | 45(14.56) | 119(38.51) |
| | Attend formal education | 29(15.26) | 161(52.10) | 19061.48) |
| Maternal level of education | Primary (1–8) | 7(3.68) | 49(25.79) | 56(29.47) |
| | Secondary (9–12) | 8(4.21) | 81(42.63) | 89(46.84) |
| | Technical /vocational | 8(4.21) | 12(6.32) | 20(10.53) |
| | Higher education (12+ | 6(3.16) | 19(10.00) | 25(13.16) |
| Residency | Rural | 67(21.68) | 46(14.88) | 113(36.56) |
| | Urban | 36(11.65) | 160(51.77) | 196(63.43) |
| Occupation | Government employer | 23(7.44) | 18(5.82) | 41(13.26) |
| | Private employer | 17(5.50) | 22(7.11) | 3912.62) |
| | Farmer | 23(7.44) | 35(11.32) | 58(18.77) |
| | Merchant | 26(8.41) | 50(16.18) | 56(24.59) |
| | Housewife | 14(5.53) | 81(26.21) | 95(30.74) |
| Current marital status | Married | 94(30.42) | 196(63.43) | 290(93.85) |
| | Other marital status** | 9(2.91) | 10(3.23) | 19(6.15) |
| Husband educational status | No attend formal school/education | 37(11.97) | 54(17.48) | 91(29.44) |
| | Attend formal school | 66(21.36) | 152(49.19) | 218(70.55) |
| Husband level of education | Primary (1–8) | 43(19.72) | 61(27.98) | 104(47.71) |
| | Secondary (9–12) | 9(4.13) | 54(24.77) | 63(28.90) |
| | Technical /vocational | 6(2.75) | 14(6.42) | 20(9.17) |
| | Higher education (12+ | 8(3.67) | 23(10.55) | 31(14.22) |
| Family size | <5 | 85(27.51) | 179(57.93) | 264(85.43) |
| | ≥5 | 18(5.83) | 27(8.74) | 45(14.56) |
| Monthly income | < 3500 | 25(8.09) | 40(12.94) | 65(48.22) |
| | 3500–9000 | 41 (13.27) | 108(34.95) | 149(48.22) |
| | ≥9000 | 37(11.97) | 58(18.77) | 95(30.74) |

NB** = marital status of mothers like divorced, widowed, single and others.

pregnancy. Additionally, 55.33% of cases had no alcohol drinking history, whereas 68.93% of controls had no history of alcohol consumption.

## Determinants of low birth weight

In the bivariable binary logistic regression analysis, several factors including sex of neonates, inadequate minimum dietary diversity of pregnant mothers, indoor air pollution, maternal educational status, types of pregnancy, lack of ANC visit, maternal age ≤20 years, maternal weight gain during pregnancy, undernutrition (MUAC ≤23cm), alcohol consumption during pregnancy, drinking coffee greater than two times per day, rural residency, hemoglobin <11gm/dl,

**Table 2. Obstetric/ Reproductive and medical related characteristics of mothers who gave birth at governmental public hospitals of north Shewa zone, Amhara regional state, Ethiopia 2023.**

| Variables | Category | Birth weight | | |
|---|---|---|---|---|
| | | Cases(n = 103) Frequency (%) | Controls(n = 206) Frequency (%) | Total (%) |
| Sex of new born | Male | 36(36.65) | 151(48.87) | 187(60.52) |
| | Female | 67(21.68) | 55(17.80) | 122(39.48) |
| Pregnancy induced hypertension | No | 78(25.24) | 167(54.05) | 245(79.29) |
| | Yes | 25(8.09) | 39(12.62) | 64(20.71) |
| PROM | No | 90(29.13) | 185(59.87) | 275(89.00) |
| | Yes | 13(4.21) | 21(6.80) | 34(11.00) |
| History of abortion | No | 96(31.07) | 196(63.43) | 292(94.50) |
| | Yes | 7(2,27) | 10(3.24) | 17(5.50) |
| Age of mother at 1st Pregnancy | < 18 | 15(4.85) | 23(7.44) | 38(12.30) |
| | ≥18 | 88(28.48) | 183(59.22) | 271(87.70) |
| Types of pregnancy | Unplanned | 63(20.38) | 46(14.89) | 109(35.27) |
| | Planned | 40(12.94) | 160(51.78) | 200(64.72) |
| History of still birth | No | 95(30.74) | 196(63.43) | 291(94.17) |
| | Yes | 8(2.59) | 10(3.24) | 18(5.83) |

and maternal healthcare decision-making autonomy were identified as significant determinants of low birth weight (LBW) with a p-value < 0.25. These variables were subsequently included in the multivariable binary logistic regression model.

Furthermore, the study investigated various factors associated with low birth weight at a significance level of p < 0.05. Among newborns, females exhibited a higher risk, with an adjusted odds ratio (AOR) of 3.13 (95% CI: 1.34, 7.32, p = 0.008), while mothers aged 20 or younger had an AOR of 3.42 (95% CI: 1.35, 8.66, p = 0.009). Notably, individuals with no formal education faced a substantially increased risk (AOR: 6.82, 95% CI: 2.94, 15.3, p < 0.001). Additionally, unplanned pregnancies were associated with a higher risk (AOR: 3.08, 95% CI: 1.38, 6.84, p = 0.006), as were instances where mothers did not attend antenatal care (AOR: 2.74, 95% CI: 1.16, 6.49, p = 0.021). Conversely, factors such as residency type and maternal age above 35 did not demonstrate significant associations (Table 4).

## Discussion

Low birth weight is still a significant cause of morbidity and mortality among neonates. This study has tried to assess determinants of LBW among neonates delivered at governmental public hospitals of North Shewa zone. Results of this study found that maternal age ≤20 years was a significant predictor of low birth weight. Those neonates from mothers of under the age of 20 had a higher risk of having a LBW (AOR: 3.42, 95% CI: 1.35, 8.66) than neonates from those mothers age between 21 to 34 years old. This result is in line with studies conducted in Mekele city and Malawi [15,16]. This similarity may be due to the sharing of similar techniques of study and lifestyle of the participants and Adolescent mothers (aged 10–19 years) face higher risks of eclampsia, puerperal endometritis, and systemic infections than women aged 20–24 years, and babies of adolescent mothers face higher risks of low birth weight, preterm birth, and severe neonatal condition [17]. But this study contradicts with study done in Nepal, which revealed no statistical association between maternal age and low birth weight [18]. This discrepancy might be due to the difference in socio-demographic characteristics of study

**Table 3. ANC service utilization and nutrition related characteristics of study participants who gave birth at governmental public hospitals of north Shewa zone, Amhara regional state, Ethiopia, 2023.**

| Variables | Category | Birth weight | | |
|---|---|---|---|---|
| | | Cases (n = 103) Frequency (%) | Controls (n = 206) Frequency (%) | Total (%) |
| ANC visit | No | 68 (22.01) | 72 (23.30) | 140(45.30) |
| | Yes | 35 (11.32) | 134(43.36) | 169(54.69) |
| Numbers of ANC visit | No ANC | 63(20.38) | 71(22.98) | 134(43.36) |
| | $\leq 3$ | 24(7.76) | 19(6,14) | 59(24.38) |
| | $\geq 4$ | 11(3.55) | 116(37.54) | 183(75.62) |
| Month at 1st ANC | 1–3 | 12(6.94) | 72 (41.62) | 84(48.55) |
| | 4–6 | 16(6.25) | 62(35.84) | 78(45.09) |
| | 7–9 | 7(4.05) | 4(2.31) | 11(6.36) |
| IFA supplementation | No | 29(9.39) | 52(16.83) | 81(26.21) |
| | Yes | 74(23.95) | 154(49.84) | 228(73.79) |
| TT injection | No | 75(24.27) | 135(43.69) | 210(67.96) |
| | Yes | 29(9.06) | 71(22.98) | 99(32.04) |
| Nutritional advice | No | 59(19.09) | 66(21.36) | 125(40.45) |
| | Yes | 44(14.24) | 140(45.31) | 184(59.55) |
| MDDS-W | Inadequate | 44(14.24) | 53(17.15) | (31.39) |
| | Adequate | 59(19.09) | 153(49.51) | 212(68.60) |
| MUAC in cm | $\leq$23cm | 55(17.80) | 37(11.97) | 92(29.77) |
| | >23cm | 48(17.53) | 169(54.69) | 217(70.23) |
| Maternal height in cm | $\leq$150cm | 16(5.18) | 29(9.39) | 45(14.56) |
| | >150cm | 87(28.16) | 177(57.28) | 264(85.44) |
| Weight gain during pregnancy | <12kg | 58(18.77) | 95(30.74) | 153(49.51) |
| | $\geq$12kg | 45(14.56) | 111(35.92) | 156(50.49) |
| Hemoglobin in mg/dl | <11mg/dl | 34(11.00) | 35(11.33) | 69(22.33) |
| | $\geq$11mg/dl | 69(22.33) | 171(55.34) | 240(77.67) |

participants and in the techniques. The other study in Finland also showed that maternal age above 40 was not increased the risk of LBW which supports this study [19].

In this study, we have examined how socio-demographic, fetal, and maternal factors predict low birth weights of newborn babies. This study showed that mothers who had given a female neonate had three times risk of LBW than those who gave a male baby (AOR: 3.13, 95% CI: 1.34, 7.32). Some studies found similar results where males were found to weigh heavier than girls at birth [10]. They explained that paternal birth weight significantly influenced the weight of boys but not for girls and further suggested that there is a genetic regulation along the male line [10].

Educational status of mothers was another significant factor that affects birth weight of baby under this study. Mothers who had no formal school attendant had nearly seven times likely to have LBW baby (AOR: 6.82, 95% CI: 2.94, 15.83). This result was supported by study done in Slovak [20] and study done in Nigeria, mothers who attained a minimum of primary, secondary, and higher education are 88.00%, 82.00%, and 57.90% likely to have NBW babies, respectively, when compared to those with no formal education [21]. This is due to the fact that education is power to solve many problems, and educated mothers had good practice of food diversity during pregnancy [22].

Unwanted and unplanned pregnancy, low ANC visits, lack of iron and folic acid supplements, and low maternal hemoglobin level were the significant determinants of low birth

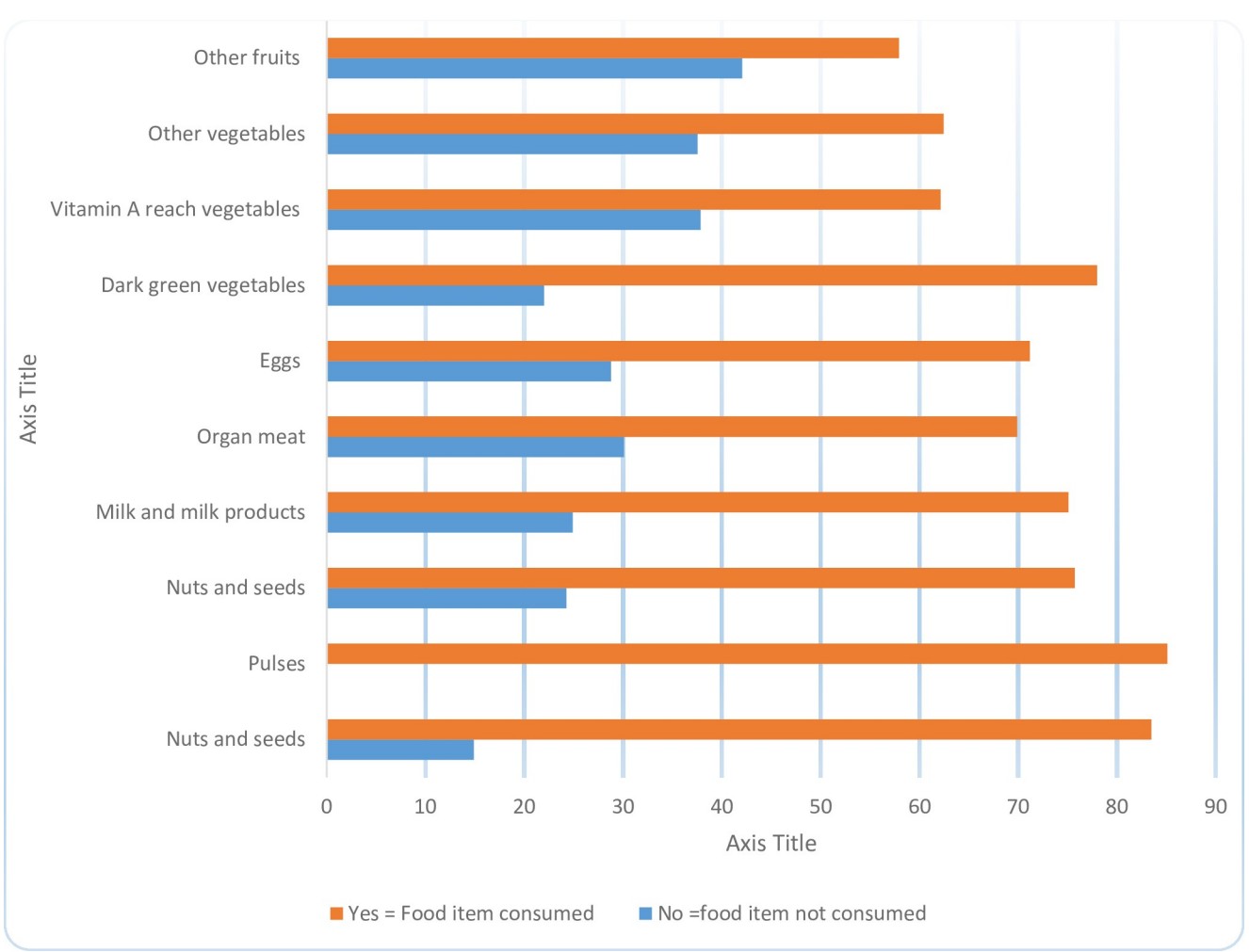

**Fig 1. Distribution of the consumption rate of ten food groups between mothers with and without low birth weight child on a daily basis, in north Shewa zone, Amhara regional state, Ethiopia, 2023.**

weight among term babies [23]. In this study, a pregnant mother who had unplanned pregnancy had three times higher risk of delivering LBW baby than who had planned pregnancy (AOR: 3.08, 95% CI: 1.38, 6.84). This is in line with a study conducted in Mekele [16] and it is due to the fact that the pregnancy is unplanned in nature, the mothers not seek health services early during pregnancy, and their health-seeking behavior is poor. In addition to this, women with unintended pregnancies were at higher risk of developing high blood pressure and anemia during pregnancy [24].

In addition, one or more antenatal care (ANC) visits were associated with reduced odds of LBW compared to those who had no follow-ups. This could be due to the fact that ANC has packages like nutritional counseling and iron supplementations for maternal and fetal wellbeing [25]. This study also revealed that mothers who did not follow ANC were nearly three times more likely to have low birth weight baby than mothers who have 3–4 ANC follow up (AOR: 2.74, 95% CI: 1.16, 6.49). This result is in line with studies conducted in public hospitals of Mekele city [16] and which is supported by a study done in South Africa [25]. In contrast, a case-control study in Ghana revealed that the number of prenatal care visits during pregnancy

**Table 4. Determine factors of low birth weight, in selected governmental public hospitals of North Shewa Zone, Amhara Regional State, Ethiopia2023.** (N = 309, case = 103 & control = 206).

| Variables | Category | Cases (n (%)) | Controls (n (%)) | AOR(95%CI) | P-value |
|---|---|---|---|---|---|
| Sex of new born | Male | 36(11.65) | 151(48.87) | | |
| | Female | 67(21.68) | 55(17.80) | 3.13(1.34,7.32) | 0.008* |
| Age of mother | ≤ 20 | 41(13.27) | 23(7.44) | 3.42(1.35,8.66) | 0.009* |
| | 21–34 | 49(15.86) | 154(49.84) | | |
| | ≥35 | 13(4.21) | 29(9.39) | 0.72(0.23,2.24) | 0.58 |
| Educational status school attendance | No | 74(23.95) | 45(14.56) | 6.82(2.94,15.3) | 0.000* |
| | Yes | 29(9.39) | 161(52.10) | | |
| Residency | Rural | 67(21.68) | 46(14.89) | 1.75(0.77,3.96) | 0.175 |
| | Urban | 36(11.65) | 160(51.78) | | |
| Types of pregnancy | Unplanned | 63(61.16%) | 46(21.35%) | 3.08(1.38,6.84) | 0.006* |
| | Planned | 40(38.83%) | 160(77.66%) | | |
| ANC visit | No | 68(22.01) | 72(23.30) | 2.74(1.16,6.49) | 0.021* |
| | Yes | 35(11.33) | 134(43.37) | | |
| MDDS-W | Inadequate | 44(14.24) | 53(17.15) | 2.67(1.16,6.13) | 0.020* |
| | Adequate | 59(19.09) | 153(49.51) | | |
| Alcohol consumption | No | 57(18.45) | 142(45.95) | | |
| | Yes | 46(14.89) | 64(20.71) | 0.94 (0.40,2.24) | 0.090 |
| Indoor cooking | No | 39(12.62) | 147(47.57) | | |
| | Yes | 64(20.71) | 59(19.09) | 3.89 (1.67,3.05) | 0.002* |
| Drinking coffee | No | 37(11.97) | 134(43.37) | | |
| | Yes | 66(21.36) | 72(23.30) | 2.02(0.83,4.89) | 0.118* |
| MUAC | ≤23 | 55(29.80) | 37(11.97) | 5.23(3,09,8.85) | 0.000* |
| | >23 | 48(15.53) | 169(54.69) | 1 | |

was negatively associated with PB and LBW but not with SGA outcome [26]. This discrepancy might be due to a small study setting.

Inadequate dietary diversity and under nutrition during pregnancy independently and significantly affected LBW in this study. Mothers having inadequate MDD-W had significantly higher odds of giving birth to LBW babies (AOR: 2.67, 95% CI 1.16, and 6.13). This finding was consistent with a study in central India that showed a 20% lesser chance of delivering LBW child with increasing maternal dietary diversity scores, in contrast, poor maternal diet quality during pregnancy may result in adverse birth outcomes with long-term.

## Conclusion and recommendations

The study highlights the urgent need to address LBW in North Shewa Zone, Amhara Region, Ethiopia, where determinants like maternal age ≤ 20 years and inadequate dietary diversity pose significant risks. Enhanced education, strengthened antenatal care, nutritional support, and community interventions are recommended. Policy initiatives advocating for improved maternal education and healthcare access are vital. Implementing these strategies will mitigate LBW incidence and improve maternal and child health outcomes in the region.

## Supporting information

**S1 File.**
(XLSX)

## Author Contributions

**Conceptualization:** Beniyas Minda, Girma Bekele, Solomon Hailemeskel, Abera Lambebo.

**Data curation:** Beniyas Minda, Girma Bekele.

**Formal analysis:** Beniyas Minda, Girma Bekele, Solomon Hailemeskel, Abera Lambebo.

**Funding acquisition:** Beniyas Minda, Girma Bekele.

**Investigation:** Beniyas Minda, Girma Bekele, Solomon Hailemeskel, Abera Lambebo.

**Methodology:** Beniyas Minda, Girma Bekele, Solomon Hailemeskel, Abera Lambebo.

**Project administration:** Beniyas Minda, Girma Bekele, Solomon Hailemeskel, Abera Lambebo.

**Resources:** Beniyas Minda, Solomon Hailemeskel, Abera Lambebo.

**Software:** Beniyas Minda, Girma Bekele, Abera Lambebo.

**Supervision:** Beniyas Minda, Girma Bekele, Solomon Hailemeskel, Abera Lambebo.

**Validation:** Beniyas Minda, Girma Bekele, Solomon Hailemeskel, Abera Lambebo.

**Visualization:** Beniyas Minda, Girma Bekele, Solomon Hailemeskel, Abera Lambebo.

**Writing – original draft:** Beniyas Minda, Girma Bekele, Solomon Hailemeskel, Abera Lambebo.

**Writing – review & editing:** Solomon Hailemeskel, Abera Lambebo.

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
