## [Decision Letter · Decision Letter 0]

26 Dec 2023

PONE-D-23-36436Determinants of Low Birth Weight among Newborns Delivered in Public Hospitals of North Shewa Zone, Amhara Region, Ethiopia: A Case-Control Study (2023)PLOS ONE

Dear Dr. Lambebo,

Thank you for submitting your manuscript to PLOS ONE. After careful consideration, we feel that it has merit but does not fully meet PLOS ONE’s publication criteria as it currently stands. Therefore, we invite you to submit a revised version of the manuscript that addresses the points raised during the review process.

**As per the reviewers' comments include at the end of this email, an extensive revision "major revision" was required. **

**Please revise the introduction section as commented by Reviewer 1. Please try to synthesize and summarize in too few paragraphs. It has to be shortened but brief, precise, and comprehensive enough considering the available literatures. **

**The methods section needs much revision as detail description (should be comprehensive but summary) of the methods used is very important to replicate the study elsewhere. Ambiguities and unclear concepts should be clarified. For detail refer to the reviewers' comments. **

**As indicated by a reviewer, the statistical analysis has not been rigorously performed. Please clarify this. Please describe the results in detail, particularly the determinants of LBW, especially results from multivariable logistic regression model. **

**As discussion is not only comparing your main findings with that of other previously conducted studies, please try to include the implication of the main findings in addition to explaining the reason of the gaps between discrepancies of the findings (i.e. for lower and higher results).**

**As conclusion section was missed, please add the conclusive statements summarizing both conclusion and recommendation together. **

We look forward to receiving your revised manuscript.

Kind regards,

Takele Gezahegn Demie, MPH

Academic Editor

PLOS ONE

Journal Requirements:

- http://dx.doi.org/10.1371/journal.pone.0269479

- https://doi.org/10.1186/s13052-020-00890-9

- https://doi.org/10.1371/journal.pone.0270002

In your revision ensure you cite all your sources (including your own works), and quote or rephrase any duplicated text outside the methods section. Further consideration is dependent on these concerns being addressed.

4. Please include your tables as part of your main manuscript and remove the individual files. Please note that supplementary tables (should remain/ be uploaded) as separate "supporting information" files

Additional Editor Comments:

Dear author,

Thank you for submitting your manuscript to PLOS ONE journal.

Reviewers decide to revise your paper. The decision is 'Major revision".

Please work on the manuscript as per the reviewers' comment and resubmit the revised manuscript as sun as possible.

Reviewers' comments:

Reviewer's Responses to Questions

**Comments to the Author**

1. Is the manuscript technically sound, and do the data support the conclusions?

Reviewer #1: Partly

Reviewer #2: Yes

2. Has the statistical analysis been performed appropriately and rigorously? 

Reviewer #1: No

Reviewer #2: No

3. Have the authors made all data underlying the findings in their manuscript fully available?

Reviewer #1: Yes

Reviewer #2: No

4. Is the manuscript presented in an intelligible fashion and written in standard English?

Reviewer #1: Yes

Reviewer #2: Yes

5. Review Comments to the Author

Reviewer #1: Background

-You can merge the 2nd and the 3rd paragraph because they both are talking about the magnitude of LBW.

Generally, you described background information for manuscript in more than 10 paragraphs (some of them are incomplete with two or three sentences). Please try to synthesize and summarize in four paragraphs.

-Please bring sentences talking about the significance of your study together, than talking about significance of the study in many places.

Method section:

-You said, four hospitals were selected from 11 hospitals. But it looks that the selected hospitals are more than 4 (1 comprehensive, 1 teaching, 2 general, and the rest primary) from the following statement taken from your manuscript: The selected hospitals included one comprehensive specialized governmental hospital, one teaching hospital, two general hospitals, and the rest being primary hospitals.

-Please explain if you have any justification for including only term babies or why preterm excluded?

-One of your inclusion criteria for controls says term singleton live-births with a birth weight >2500g. What if birth weight is greater than 4000g?

-Please do not mention preterm as an exclusion criterion because it is by default excluded in your inclusion criteria

-Please explain what does new-borns with multiple births mean (it is one of your exclusion criteria)

-Please provide any justification for excluding diabetic mothers.

-Please provide more clarification on how you selected controls using systematic random sampling?

-You have stated about informed consent under data collection method. Please take it to the ethics section.

-Please add statistical analysis methods you went through

Result

-Please add reports related to determinants of LBW, especially results from multivariable logistic regression model

Please add conclusion

Reviewer #2: Comments

*Way of sample size determination is not clear, make it clear.

*You used P-Value 0.25 for binary logistic regression, justify it how it can be.

* There are no variables determined under result part. Which are significantly associated with dependent variables after entered into Multivariable logistic regression model? Put such variables in result part and explain them.

*Discussion, there is visible comparison between the current and previous study and also explain the reason of the gaps between current and previous study.

*There is no conclusion without the variable's determination on last (multivariable logistic regression), so before conclusion you should identify the variables significantly associated by multivariable logistic regression.

6. PLOS authors have the option to publish the peer review history of their article (what does this mean?). If published, this will include your full peer review and any attached files.

Reviewer #1: **Yes: **Robera Demissie Berhanu

Reviewer #2: No

---

## [Author Response · Author response to Decision Letter 0]

5 Jan 2024

I would like to express my sincere appreciation for the valuable feedback provided by the reviewers and editors. Their insightful comments and constructive criticisms have significantly contributed to improving the overall quality and clarity of the manuscript. I have thoroughly considered each comment and made the necessary revisions to address all concerns raised during the peer review process.

In the revised manuscript, you will find detailed responses to each comment along with a marked-up version highlighting the changes made. I believe that these revisions have strengthened the scientific rigor and presentation of the work.

I am confident that the revised manuscript is now ready for publication in PLOS ONE. I appreciate the thorough review process and the commitment to maintaining the high standards of the journal.

Thank you for considering my work for publication in PLOS ONE. I look forward to your favorable response.

---

## [Decision Letter · Decision Letter 1]

12 Feb 2024

PONE-D-23-36436R1Determinants of Low Birth Weight among Newborns Delivered in Public Hospitals of North Shewa Zone, Amhara Region, Ethiopia: A Case-Control Study (2023)PLOS ONE

Dear Dr. Lambebo,

Thank you for submitting your manuscript to PLOS ONE. After careful consideration, we feel that it has merit but does not fully meet PLOS ONE’s publication criteria as it currently stands. Therefore, we invite you to submit a revised version of the manuscript that addresses the points raised during the review process.

We look forward to receiving your revised manuscript.

Kind regards,

Takele Gezahegn Demie, MPH

Academic Editor

PLOS ONE

Journal Requirements:

Reviewers' comments:

Reviewer's Responses to Questions

**Comments to the Author**

1. If the authors have adequately addressed your comments raised in a previous round of review and you feel that this manuscript is now acceptable for publication, you may indicate that here to bypass the “Comments to the Author” section, enter your conflict of interest statement in the “Confidential to Editor” section, and submit your "Accept" recommendation.

Reviewer #2: (No Response)

2. Is the manuscript technically sound, and do the data support the conclusions?

Reviewer #2: No

3. Has the statistical analysis been performed appropriately and rigorously? 

Reviewer #2: Yes

4. Have the authors made all data underlying the findings in their manuscript fully available?

Reviewer #2: No

5. Is the manuscript presented in an intelligible fashion and written in standard English?

Reviewer #2: Yes

6. Review Comments to the Author

Reviewer #2: Still the majority of comments were not corrected and responded as they were commented.

*You used P-Value 0.25 for binary logistic regression, justify it how it can be.

* There are no variables determined under result part. Which are significantly associated with dependent variables after entered into Multivariable logistic regression model? Put such variables in result part and explain them.

* There is no conclusion without the variable's determination on last (multivariable logistic regression), so before conclusion you should identify the variables significantly associated by multivariable logistic regression.

7. PLOS authors have the option to publish the peer review history of their article (what does this mean?). If published, this will include your full peer review and any attached files.

Reviewer #2: No

---

## [Editor Report · Decision Letter 2]

12 Mar 2024

PONE-D-23-36436R2Determinants of Low Birth Weight among Newborns Delivered in Public Hospitals of North Shewa Zone, Amhara Region, Ethiopia: A Case-Control Study (2023)PLOS ONE

Dear Dr. Lambebo,

Thank you for submitting your manuscript to PLOS ONE. After careful consideration, we feel that it has merit but does not fully meet PLOS ONE’s publication criteria as it currently stands. Therefore, we invite you to submit a revised version of the manuscript that addresses the points raised during the review process.

We look forward to receiving your revised manuscript.

Kind regards,

Takele Gezahegn Demie, MPH

Academic Editor

PLOS ONE

Journal Requirements:

Additional Editor Comments:

Dear Author,

Thank you for resubmitting the revised version of your manuscript.

Please include detailed point-by-point response in writing to reviewer's comments/suggestions rather than writing "accepted and corrected'. Even though you indicated the changes made in yellow color, you have to discuss Wwhere within the manuscript and what changes have been made. This should be attached. While doing so, please describe anything which needs clarification in detail and justify points that need justification. If your response is negative, that should also be convincing to the reviewer as well as the prospective readers in case this paper is accepted for publication.

---

## [Author Response · Author response to Decision Letter 2]

20 Mar 2024

PONE-D-23-36436R2

Determinants of Low Birth Weight among Newborns Delivered in Public Hospitals of North Shewa Zone, Amhara Region, Ethiopia: A Case-Control Study (2023)

PLOS ONE

Dear Dr. Lambebo,

Thank you for submitting your manuscript to PLOS ONE. After careful consideration, we feel that it has merit but does not fully meet PLOS ONE’s publication criteria as it currently stands. Therefore, we invite you to submit a revised version of the manuscript that addresses the points raised during the review process.

We look forward to receiving your revised manuscript.

Kind regards,

Takele Gezahegn Demie, MPH

Academic Editor

PLOS ONE
---

## [Editor Report · Decision Letter 3]

24 Apr 2024

Determinants of Low Birth Weight among Newborns Delivered in Public Hospitals of North Shewa Zone, Amhara Region, Ethiopia: A Case-Control Study (2023)

PONE-D-23-36436R3

Dear Dr. Lambebo,

We’re pleased to inform you that your manuscript has been judged scientifically suitable for publication and will be formally accepted for publication once it meets all outstanding technical requirements.

Kind regards,

Takele Gezahegn Demie, MPH

Academic Editor

PLOS ONE
---

## [Editor Report · Acceptance letter]

30 Apr 2024

PONE-D-23-36436R3 

PLOS ONE

Dear Dr. Lambebo, 

I'm pleased to inform you that your manuscript has been deemed suitable for publication in PLOS ONE. Congratulations! Your manuscript is now being handed over to our production team.

Kind regards, 

on behalf of

Mr. Takele Gezahegn Demie 

Academic Editor

PLOS ONE